# MIF Variant rs755622 Is Associated with Severe Crohn’s Disease and Better Response to Anti-TNF Adalimumab Therapy

**DOI:** 10.3390/genes14020452

**Published:** 2023-02-09

**Authors:** Gregor Jezernik, Mario Gorenjak, Uroš Potočnik

**Affiliations:** 1Faculty of Medicine, University of Maribor, Taborska Ulica 8, 2000 Maribor, Slovenia; 2Faculty of Chemistry and Chemical Engineering, University of Maribor, Smetanova Ulica 17, 2000 Maribor, Slovenia; 3Department for Science and Research, University Medical Centre Maribor, Ljubljanska Ulica 5, 2000 Maribor, Slovenia

**Keywords:** treatment outcome, infliximab, adalimumab, biomarkers, Crohn’s disease, rheumatoid arthritis, psoriatic arthritis, ankylosing spondylitis

## Abstract

Crohn’s disease (CD), rheumatoid arthritis, psoriatic arthritis and other inflammatory diseases comprise a group of chronic diseases with immune-mediated pathogenesis which share common pathological pathways, as well as treatment strategies including anti-TNF biologic therapy. However, the response rate to anti-TNF therapy among those diseases varies, and approximately one third of patients do not respond. Since pharmacogenetic studies for anti-TNF therapy have been more frequent for other related diseases and are rare in CD, the aim of our study was to further explore markers associated with anti-TNF response in other inflammatory diseases in Slovenian CD patients treated with the anti-TNF drug adalimumab (ADA). We enrolled 102 CD patients on ADA, for which the response was defined after 4, 12, 20 and 30 weeks of treatment, using an IBDQ questionnaire and blood CRP value. We genotyped 41 SNPs significantly associated with response to anti-TNF treatment in other diseases. We found novel pharmacogenetic association between SNP rs755622 in the gene *MIF* (macrophage migration inhibitory factor) and SNP rs3740691 in the gene *ARFGAP2* in CD patients treated with ADA. The strongest and most consistent association with treatment response was found for the variant rs2275913 in gene *IL17A* (*p* = 9.73 × 10^−3^).

## 1. Introduction

Crohn’s disease (CD), rheumatoid arthritis (RA), psoriatic arthritis (PA) and other inflammatory diseases comprise a group of chronic diseases with immune-mediated pathogenesis. Since these are all inflammatory diseases, they share common pathological pathways [1]. Tumor necrosis factor α inhibitors (anti-TNF) have improved the treatment of the majority of autoimmune inflammatory complex diseases, and give substantial improvement in cases where convenient treatment using non-steroidal anti-inflammatory drugs (NSAIDs), corticosteroids and antibiotics was not successful. Targeting tumor necrosis factor α in IBD allows intestinal healing by blocking TNFR1-dependant intestinal epithelial cell death [2] and inducing cell death in macrophages by binding to transmembrane TNF or by depriving TNFR2-dependent CD4+ T cell survival via NF-κB activation [3]. However, 30–40% of patients have an inadequate response to anti-TNF drugs. It is known that genetic factors influence the response to anti-TNF treatment [4]. Anti-TNF pharmacogenetic studies, including the different anti-TNF drugs infliximab (IFX), adalimumab (ADA) and etanercept (ETN), have so far been performed in RA, inflammatory bowel disease (IBD), PA and ankylosing spondylitis (AS) patients. Most of them have focused on candidate genes known to play a role in susceptibility to disease, and genes implicated in TNFα signaling pathways. Recently, some genome wide association studies (GWAs) of genetic predictors of anti-TNF treatment efficacy have been also performed in RA [5,6] and pediatric IBD [4]. Pharmacogenomic GWA studies are performed on modest sample sizes compared to genetic studies of disease risk. Furthermore, there is a considerable difference between the type of anti-TNF treatment [7] and use of concomitant medications [8]. Moreover, the advent of biosimilars has lowered the treatment cost and increased the drug supply, thereby bringing anti-TNF treatment to even more patients [9]. That is why independent cohort studies are required to validate findings from other cohorts, diseases and GWAs further. The aim of our study is the exploration of previously published anti-TNF response markers in our well-defined Slovenian CD patient cohort treated with ADA.

## 2. Materials and Methods

### 2.1. Literature Search and SNP Selection

An extensive literature search was performed in PubMed using various search terms and combinations of these terms, such as anti-TNF, pharmacogenetic study, genome-wide association study, autoimmune diseases, etc. Pharmacogenetic studies of anti-TNF response in autoimmune inflammatory diseases, including Crohn’s disease (CD), ulcerative colitis (UC), RA, PA, AS, spondyloarthritis (SpA) and multiple sclerosis (MS), published between the years 2001 and 2015, were used to identify loci associated with response to biological therapy using anti-TNF agents, including IFX, ADA or ETN. All SNPs that were associated with response to any anti-TNF drug were included.

### 2.2. Patients

We enrolled patients with CD on ADA as described previously [10]. Exclusion criteria were: other complications of CD (e.g., stenosis, abscesses, previous total colectomy), a history of allergy to murine proteins, a serious infection in the previous 3 months, positive test on tuberculosis or active tuberculosis, malignancy, pregnancy and lactation) [10]. Briefly, 102 Slovenian patients with refractory CD were investigated over a period of 30 weeks. Before the first dose of ADA and 4, 12, 20 and 30 weeks after treatment, the response was determined using an IBDQ questionnaire score (clinical response) and blood CRP value (biological response). Both the clinical and biological responses were determined as the difference in the IBDQ or CRP value before and after treatment. Clinical response was defined as an increase in IBDQ by more than 22 points (ΔIBDQ > 22), or as an IBDQ value higher than 170 points [11], and biological response as a decrease in CRP to normal values (<3 mg/L), or a drop in CRP levels by more than 25% [12,13]. The clinical characteristics (including disease location and behavior according to the Montreal classification) of the cohorts are listed in Table 1.

### 2.3. Genetic Analysis

DNA samples were available for all patients. The DNA was isolated from peripheral blood lymphocytes using a TRI reagent (Sigma, Darmstadt, Germany) according to the manufacturer’s instructions. Genotypes for forty-one (41) SNPs selected from anti-TNF pharmacogenetic studies performed in PA, RA, IBD, CD, UC, MS, AS and SpA were extracted from our genotype data bank. The genotype data bank was obtained using the iChip platform as described previously [14].

### 2.4. Gene Ontology Analysis

Functional annotation was performed using publicly available functional and biological databases. Gene ontology analysis was performed using the software package CytoScape 3.8.2. [15] with the integrated application ClueGO v2.5.8. [16]. A ClueGO analysis was performed using the following parameters and selected options:Ontology/Pathways selected:Biological Process (13.05.2021)Cellular Component (13.05.2021)Molecular Function (13.05.2021)Evidence selected: only *All_Experimental*

Statistical significance was defined as a *p* value lower than 5 × 10^−2^ after Bonferroni step-down correction (the default selection in ClueGO v2.5.8).

To enhance biological process discovery with gene ontology analysis, the lists of investigated genes were extended to include their interactors. Genes interacting with at least two investigated genes (i.e., genes associated with response to ADA) were obtained from the BIOGRID database [17,18] using the biogridR package [19] for R 4.1.1 [20].

### 2.5. Statistical Analysis

We used the two-sided Fisher’s exact test to compare genotype and allele frequencies between response to treatment as a categorical variable (response versus non-response) to ADA treatment. To compare continuous data between different genotypes (dominant and recessive models) and alleles or treatment response, we used an independent samples t-test in cases of normal distribution of data (the Kolmogorov–Smirnov test—*p* > 0.05), or the Mann–Whitney U-test in cases where the data deviated significantly from a normal distribution (the Kolmogorov–Smirnov test of normality—*p* < 0.05). Fisher combined *p*-value analysis was performed in cases where associations had been significant according to different statistics calculated from the same sample (multi-phase analysis). For the statistical analysis we used the IBM SPSS Statistics 22.0 statistical package.

## 3. Results

### 3.1. Literature Search

Out of 22 studies we collected 73 SNPs from 47 independent loci associated with response to anti-TNF therapy. In total, 50% of the studies (11 out of 22) were performed in IBD patients (IBD pediatric, IBD adults, CD or UC), and 36.4% of the studies (8 out of 22) were performed in RA patients. In total, 64.4% (47) of SNPs were associated with response to anti-TNF therapy in a group of IBD patients, 31.5% (23) in a group of RA patients and 9.6% (7) in a group of AS patients. Two SNPs were associated with anti-TNF response in a group of patients with SpA, one SNP in patients with PA, and one SNP in patients with MS. Table 2 summarizes the pharmacogenetic signals statistically significantly associated with response to anti-TNF therapy in RA, PA, SpA, IBD (CD and UC), pediatric IBD (PED-IBD), AS and MS patients, in the years between 2001 and 2015. Additional variant information is contained within Appendix A (Appendix A).

### 3.2. Pharmacogenetic Analysis

No statistically significant associations were detected between the analyzed SNPs and clinical data. Three loci showed strong association with treatment response to ADA in CD patients. The most consistent association during 30 weeks of treatment with ADA was observed between the SNP rs2275913 in gene *IL17A* and the response measured by the IBDQ. Patients with a GG genotype of SNP rs2275913 had a better response compared to patients with an AA or AG genotype (Figure 1). The strongest statistically significant association was confirmed after 20 weeks of treatment. The average difference in the IBDQ value for patients with the genotype GG was higher (31.9) compared to patients with the AA or AG genotype (13.8, *p* = 9.73 × 10^−3^). After 20 weeks of treatment 75.5% of patients with the genotype GG had a positive response to anti-TNF therapy with ADA compared to 54.5% of patients with response with genotype AA or AG (*p* = 4.67 × 10^−2^). The same tendency was observed during all 30 weeks of treatment (Table 3). The combined *p*-value analysis showed the strongest statistical significance after 20 weeks of treatment (*p* = 6.43 × 10^−4^).

Consistent association during all 30 weeks of treatment was also observed for the SNP rs755622 in the gene *MIF* (Figure 2). Patients with a GG genotype showed better response compared to patients with a CC or CG genotype. After 4 weeks of treatment with ADA, patients with GG had a higher deltaIBQ (60.6) compared to patients with a CC or CG genotype (16.4, *p* = 4.00 × 10^−3^). The same tendency was also observed after 12, 20 and 30 weeks of treatment. Association has also been observed for biological response measured with CRP, where patients with the genotype GG had a higher deltaCRP after 12 weeks of treatment (11.3) compared to patients with a CC or CG genotype (6.8, *p* = 0.026). Interestingly, patients with the GG genotype had a significantly lower IBDQ value (122) before treatment compared to patients with CC or CG genotypes (155, *p* = 0.039).

After four weeks of treatment, a strong, statistically significant association was confirmed for SNP rs3740691 in the gene *ARFGAP2* (Figure 3). In the group of patients with genotype AA or AG there were 59.6% of nonresponders compared to 15.1% of nonresponders in the group of patients with genotype GG (*p* = 1.24 × 10^−5^). Furthermore, after four weeks of treatment, the average IBDQ value in patients with genotype AA or AG reached only 158.3 points compared to 183.7 points in patients with the genotype GG (*p* = 2.74 × 10^−4^). The difference also remained significant after 12 weeks of treatment. The combined p-value analysis showed the strongest statistical significance for SNP rs3740691 in gene *ARFGAP2* after 4 weeks of treatment (*p* = 2.24 × 10^−9^).

Significant associations were confirmed for genes involved in the regulation of NF-κB signaling, particularly *TLR2* (*p* = 1.48 × 10^−3^), *TLR4* (*p* = 1.36 × 10^−2^) and *TLR9* (*p* = 1.98 × 10^−2^).

Associations between the response to ADA therapy in Slovenian CD patients were altogether found for 28 out of the analyzed 41 SNPs. Our analysis replicated 17 (36.2%) of 47 SNPs associated with anti-TNF response in IBD. Not all SNPs could be replicated reliably due to the nature of IBD and anti-TNF response as a complex trait. The majority of the confirmed associations were between SNPs already associated with response to any anti-TNF drug in IBD (UC or CD) patients. However, the highest overlap was observed between SNPs associated with the response to anti-TNF therapy in AS patients. All associations are presented in Table 3.

### 3.3. Gene Ontology Analysis

To analyze whether there are specific processes associated with response to anti-TNF therapy with ADA in CD patients, we first performed gene ontology analysis only for genes associated with response (i.e., the genes listed in Table 3). Secondly, we extended the gene ontology analysis to interactors of genes listed in Table 3 obtained from BIOGRID. Finally, we performed gene ontology analysis for all genes reported to be associated with response to anti-TNF therapy in PA, RA, IBD, CD, UC, SpA, MS and AS from Table 2.

Gene ontology analysis of the genes listed in Table 3 showed few significant results, but their extended list containing BIOGRID interactors revealed several enriched GO terms. The genes and their interacting nodes are visualized in Figure 4. Many highly significant enriched GO terms are related to NF-kappaB signaling and TNFα, the most significant being *I-kappaB kinase/NF-kappaB signaling* (*p* = 2.76 × 10^−37^). Other significant terms include *death-inducing signaling complex assembly* (*p* = 4.67 × 10^−15^) and *TRIF-dependent Toll-like receptor signaling pathway* (*p* = 1.28 × 10^−25^).

The addition of genes from Table 2 to the gene ontology analysis did not alter or expand the GO results significantly. The newly identified leading terms are *response to bacterium* (*p* = 3.30 × 10^−12^) and related hyponyms. The full results of the gene ontology analysis are shown in Appendix A.

Moreover, pathways of interest were selected and visualized based on genes of interest and their interactors, as well as statistically significant GO results (Figure 5). Figure 5 displays primarily a part of the TNFR1-related pathways, TRIF-dependent Toll-like receptor signaling for TLR3 and TLR4. Figure 5 is based on images published by Rusu et al. [2] and Aluri et al. [39].

## 4. Discussion

In the present study we performed an extensive pharmacogenetic study in CD patients treated with ADA. This is the first replication study of cross-disease anti-TNF signals performed in a well-characterized cohort of refractory CD patients treated specifically with the anti-TNF drug ADA. We identified a novel association between the *MIF* variant rs755622, severity of CD and response to anti-TNF therapy in CD patients, and also confirmed the *IL17A* variant rs2275913 as the strongest and most consistent predictor of response to ADA in CD patients during 30 weeks of treatment. For the *IL17A* variant rs2275913 we found a better response in patients with genotype GG compared to patients with the genotype AA or AG. On average, after 20 weeks of treatment, patients with the GG genotype had a 30-point increase in IBDQ score, a 20-point increase in the IBDQ value, and among patients with the GG genotype, there were 82% of responders compared to 53% of responders with the AA or AG genotype. Our finding is consistent with the observation that IBD patients with an AG or AA genotype of *IL17A* rs2275913 SNP are associated with nonresponse [26], and is contrary to the finding that female RA patients carrying the GG genotype are characterized by a poor response to anti-TNF treatment [25]. IL-17A, which is produced mainly by Th17 cells, mediates autoimmunity and immune defense against pathogens, and is increased in the intestinal mucosa of patients affected by chronic inflammatory bowel disorders, such as celiac disease, CD, and UC. It was also found that IL-17A is overexpressed in CD strictures compared with non-structured CD areas and gut control [40], and that IL-17A is increased in the inflamed areas of patients with IBD [41], and has a role in epithelial permeability independent of IL-23 [42], further confirming the role of *IL17A* in the pathogenesis of CD. Indeed, the association with an inflated gut area and gut permeability could open the epithelium to more common host–microbe interactions in the otherwise sterile lamina propria, leading to frequent inflammatory responses and greater odds of chronic and severe disease progression. Meanwhile, *MIF* is a key cytokine in RA, and changes following anti-TNF therapy were observed in RA almost a decade ago [43]. However, similar changes have not been observed in CD.

For the first time we identified an association between SNP rs755622 in the gene *MIF* (macrophage migration inhibitory factor) and response to anti-TNF therapy using ADA in patients with CD. We found a better response in patients with the GG genotype for SNP rs755622, in which the difference in the IBDQ score was higher in the 30 week period of treatment compared to patients with a CG or CC genotype. However, patients with the genotype GG were associated with more severe disease prior to the introduction of the anti-TNF therapy. An association between SNP rs755622 and response to anti-TNF therapy has been found in RA patients where minor allele G predicted nonresponse to anti-TNF treatment [24], which is contrary to our results. However, SNP rs755622 was previously associated with more active disease in RA patients [44], where carriers of the minor allele had higher levels of circulating MIF and higher levels of radiological joint damage. Interestingly, AS patients with rs755622 risk allele G also reflect a more active disease [24]. These two observations are consistent with our finding that patients with genotype GG have more severe disease before anti-TNF therapy.

For the first time, we report the association of SNP rs3740691 in gene *ARFGAP2* with the response to ADA in CD patients.

Associations between a response to ADA therapy in Slovenian CD patients were altogether found in 28 out of the analyzed 41 SNPs, confirming high overlap with other related diseases. The majority of the confirmed associations were between SNPs already associated with response to any anti-TNF drug in IBD patients. However, the highest overlap was observed between SNPs associated with response to anti-TNF therapy in ankylosing spondylitis (AS) patients. AS is characterized by prominent inflammation of the axial skeleton, although other joints may also be affected. The relationship between IBD and AS has been known for many years. Approximately 5–10% of AS patients have concomitant IBD, either CD or UC, and first-degree relatives of patients with AS are ~3 times more likely to develop CD or UC than unrelated individuals [45,46,47]. Similarly, half of IBD patients develop chronic back pain, and this back pain progresses in about 5 to 10% of IBD patients to become a spondylopathy disorder [48]. Further, current data are consistent with the hypothesis that defective gut mucosal immunity is a major driver of AS, and many genetic associations in AS and IBD overlap [47]. Although genetic studies confirmed the association between AS and CD, the overlap between the pharmacogenetic markers has not been evaluated so far. In our study, a high overlap between markers of response to anti-TNF therapy has been found between AS and CD, further confirming cross-phenotype similarities and associations. From a therapeutic perspective, both infliximab and adalimumab are indicated for use in IBD and AS. Interestingly, another anti-TNF agent, etanercept, has been indicated for AS, but failed to achieve a therapeutic effect in IBD. The current understanding of etanercept’s failure in IBD highlights its ability to bind to soluble, but not transmembrane, TNF, which is believed to be the key source of TNF-related pathogenic effects in IBD [49].

We also wanted to explore the biological pathways and molecular functions involved in the response to anti-TNF treatment in CD, and compare them with other related common autoimmune diseases. Inflammatory response through NF-kappaB signaling has been found to be the most significant biological pathway associated with response to ADA. The results of the GO analysis related to NF-kappaB signaling highlight its importance in anti-TNF response, since NF-κB is the principal mediator of several pro-inflammatory processes in different immune-mediated diseases, including CD. Aberrant changes in genes involved in NF-κB signaling may lead to lower rates of satisfactory anti-TNF response. Additional functional testing is required to explain how variants in genes associated with GO terms related to NF-κB may affect the anti-TNF response.

The GO term result death-inducing signaling complex assembly is likely related to TNFR1 signaling, which mediates both the TNF-induced canonical NF-κB pathway and the TNFR1-dependent death induction. Anti-TNF therapy is also believed to ameliorate aberrant changes in TNFR1-dependent pathways by preventing canonical NF-κB pathway activation and cell death of affected tissue, such as the gut epithelium in IBD and the synovium in RA. In addition, the GO term TRIF-dependent Toll-like receptor signaling pathway may also be related to the described TNFR1-dependent signaling pathway. In IBD, both TLR3 and TRIF can contribute to the TNFR1-dependent pathway, which leads to cell death in the intestinal epithelium [2], but may also activate apoptosis and necroptosis pathways independent of TNFR1.

When comparing all genes associated with response to anti-TNF therapy, the response to bacterium, more specifically, the response to lipopolysaccharide, showed the strongest association, followed again by an inflammatory response through NF-kappaB signaling. So far, few studies have analyzed the biological processes involved in response/nonresponse to anti-TNF therapy. In a GWAS of anti-TNF response in RA patients strong involvement has been confirmed of the biological processes underlying the inflammatory response and cell morphology [6]. In our study, NF-kappaB signaling and bacterium response pathways are recognized as one of the most important processes contributing to a response to ADA.

## Figures and Tables

**Figure 1 genes-14-00452-f001:**
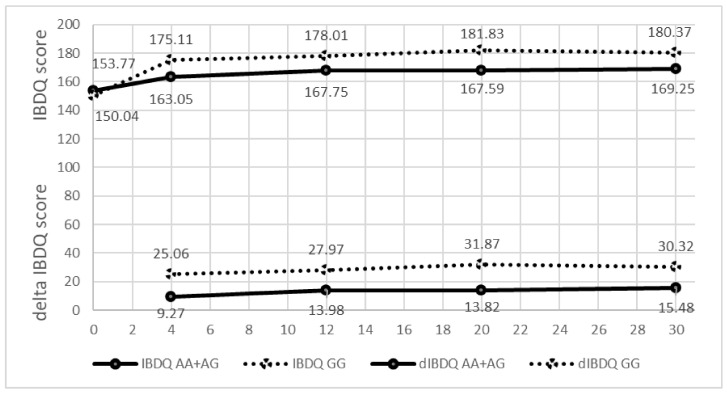
IBDQ value and difference in the IBDQ value (delta IBDQ) during the 30-week treatment period according to a genotype of SNP rs2275913 in the *IL17A* gene.

**Figure 2 genes-14-00452-f002:**
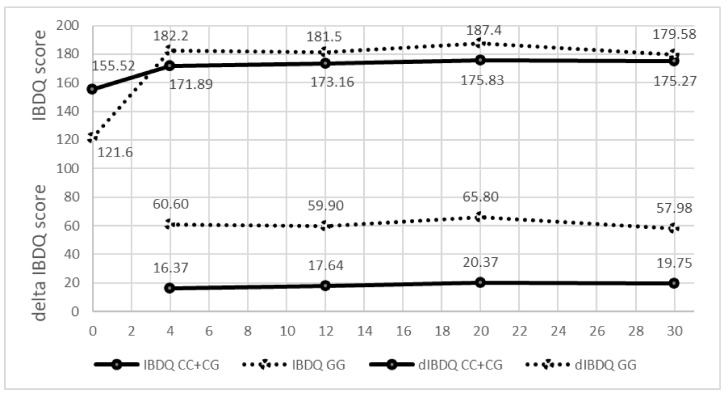
IBDQ values and the difference in the IBDQ values (delta IBDQ) during the 30-week treatment period according to the genotype of SNP rs755622 in the MIF gene.

**Figure 3 genes-14-00452-f003:**
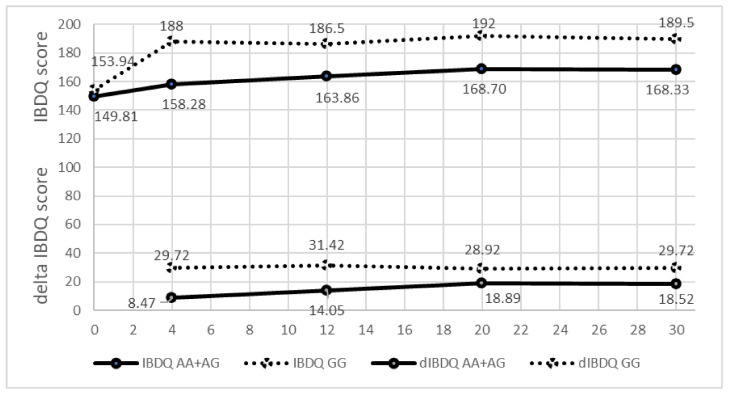
IBDQ value and the difference in the IBDQ value (delta IBDQ) during the 30-week treatment period according to the genotype of the SNP rs3740691 in the gene *ARFGAP2*.

**Figure 4 genes-14-00452-f004:**
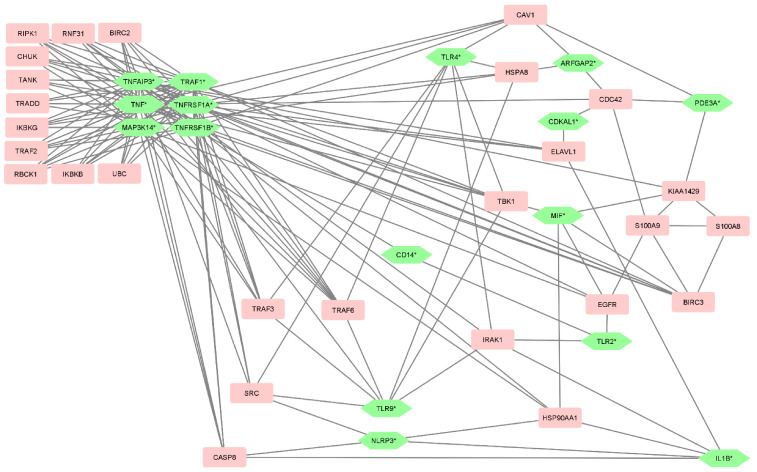
Genes of interest and their interacting genes. The green hexagonal nodes represent genes of interest, also marked with * after the gene name. The pink rectangles represent interacting genes obtained from BIOGRID.

**Figure 5 genes-14-00452-f005:**
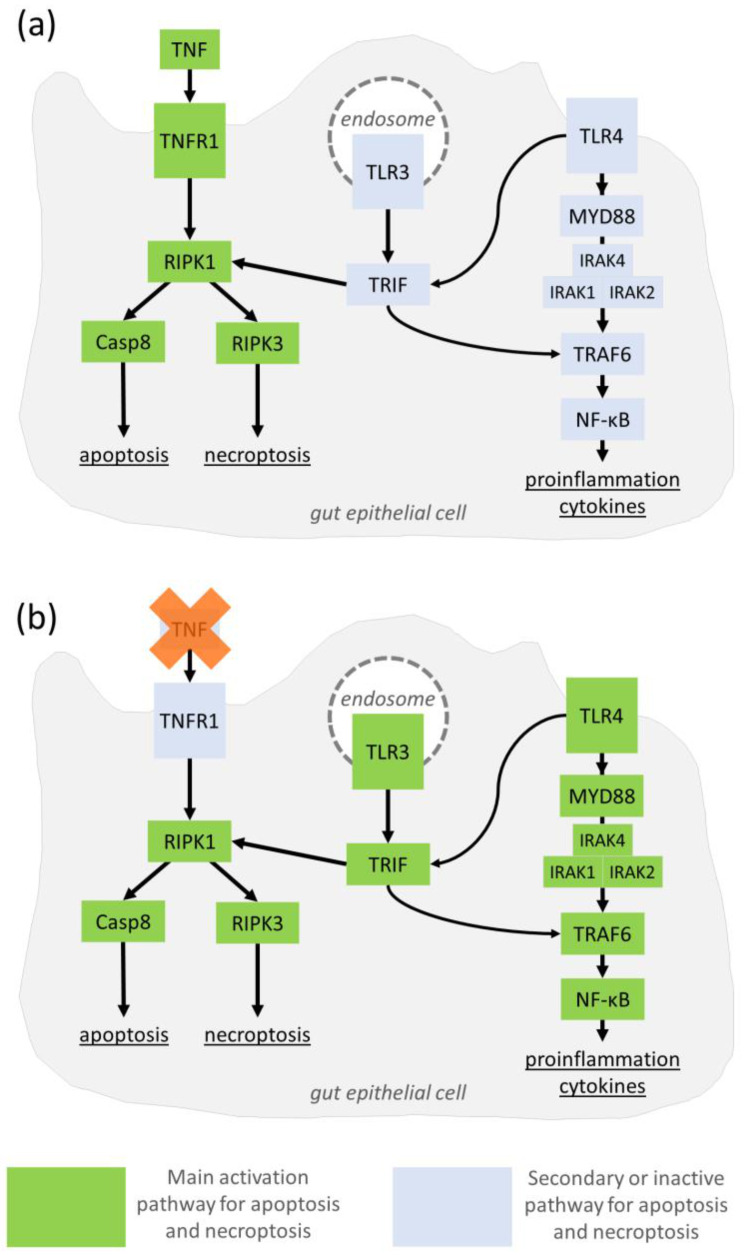
TNF-independent pathways of non-response in gut epithelial cells. Genes with variants associated with response are outlined. (**a**) In the presence of aberrant TNF signaling in IBD, the TNFR1-associated pathway will induce excessive apoptosis or necroptosis. (**b**) During anti-TNF drug therapy, apoptosis or necroptosis (but also NF-κB pathways) may still be induced independent of TNF via TRIF-dependent TLR3 (or TLR4 [39]) signaling [2]. Deleterious changes in TRIF-dependent Toll-like receptor signaling may lead to molecular pathology of CD, which is less reliant on TNF expression alone, and, thus, resistant to TNF treatment.

**Table 1 genes-14-00452-t001:** Summary of clinical data.

Number of patients		102
Average age at enrollment		41.1
Average age at diagnosis		27.4
Sex	Male	43
	Female	59
Previously required operative procedure	Yes	58
	No	44
Disease location	L1	18
	L2	32
	L3	52
	L4	2
Disease behavior	B1	35
	B2	28
	B3	35
	B2 + B3	4
	Perianal manifestations	7
Concurrent drug use	5-aminosalicylic acid	47
	Corticosteroids	36
	Azathioprine or 6-mercaptopurine	31
Smoking	Yes	38
	No	64
Average IBDQ value	Week 0	152.42
	Week 4	169.48
	Week 12	173.36
	Week 20	175.08
	Week 30	175.43
Blood CRP	Week 0	19.48
	Week 4	12.42
	Week 12	11.15
	Week 20	10.60
	Week 30	12.55

**Table 2 genes-14-00452-t002:** Summary of pharmacogenetic anti-TNF signals.

Disease ^1^	SNP ID	Chr	Gene	Study	Correlation (Genotype x Response)	*p* Value, OR
PA	rs3794271	12	PDE3A	[21]	G—nonresponse	*p* = 0.0036
CD, AS, SpA	rs1800629	6	TNF	[22]	G—response	*p* = 0.0007
CD, AS, SpA, RA	rs1799724	6	TNF	[22,23]	C—response	*p* = 0.010
CD, AS, SpA, RA	rs1799724	6	TNF	[22,23]	TT—response	*p* = 0.045
AS	rs917997	2	IL18RAP	[24]	A—nonresponse	*p* = 0.007
AS	rs755622	22	MIF	[24]	G—nonresponse	*p* = 0.019
AS	rs1800896	1	IL10	[24]	A—nonresponse	*p* = 0.041
AS	rs3740691	11	ARFGAP2	[24]	A—nonresponse	*p* = 0.002
AS	rs1061622	1	TNFRSF1B	[24]	G—nonresponse	*p* = 0.048
IBD, RA	rs2275913	6	IL17A	[25,26]	GG—nonresponse, GA or AA—nonresponse	*p* = 0.040
RA, IBD	rs2275913	6	IL17A	[25,26]	GA or AA— nonresponse	*p* = 0.050
RA, CD	rs767455	12	TNFRSF1A	[23,27]	AA—response	*p* = 0.040
RA	rs3761847	9	TRAF1	[28]	GG—nonresponse	OR = 7.4
RA	rs4612666	1	NLRP3	[29]	T—nonresponse	*p* = 0.025
IBD, RA	rs2430561	12	IFNG	[26,29]	TA or AA— response, A—nonresponse	*p* = 0.005
PED-IBD	rs2836878	21	BRWD1	[4]	GG—nonresponse	*p* = 0.030
PED-IBD	rs2188962	5	IRF1-AS1	[4]	CC—nonresponse	*p* = 0.028
PED-IBD	rs6908425	6	CDKAL1	[4]	CT or TT—nonresponse	*p* = 0.049
PED-IBD	rs2395185	6	HLA-DQA1	[4]	GG—nonresponse	*p* = 0.007
PED-IBD	rs2241880	2	ATG16L1	[4,30]	AA or AG—nonresponse	*p* = 0.048
PED-IBD	rs2241880	2	ATG16L1	[4,30]	GG—response	*p* = 0.010
PED-IBD	rs762421	21	ICOSLG	[4]	AA—nonresponse	*p* = 0.045
MS	rs1800693	12	TNFRSF1A	[31]	G—nonresponse	NA
CD	rs1816702	4	TLR2	[26]	CT or TT—response	*p* = 0.040
IBD	rs3804099	4	TLR2	[26]	CT or CC—response	*p* = 0.010
IBD	rs11938228	4	TLR2	[26]	AC or AA—nonresponse	*p* = 0.040
IBD	rs5030728	9	TLR4	[26]	AG or AA—response	*p* = 0.010
IBD	rs1554973	9	TLR4	[26]	CT or CC—nonresponse	*p* = 0.040
IBD	rs187084	3	TLR9	[26]	CT—response	*p* = 0.040
IBD	rs352139	3	TLR9	[26]	AA—nonresponse	*p* = 0.040
IBD	rs11465996	8	LY96	[26]	CG or GG—response	*p* = 0.010
IBD	rs7222094	17	MAP3K14	[26]	CT–response	*p* = 0.050
IBD	rs361525	6	TNF	[26]	AG—nonresponse	*p* = 0.040
IBD	rs4149570	12	TNFRSF1A	[26]	TT—response	*p* = 0.030
IBD	rs6927172	6	TNFAIP3	[26]	CG or GG—nonresponse	*p* = 0.030
IBD	rs4848306	2	near IL1B	[26]	AG or GG—response	*p* = 0.040
IBD	rs10499563	7	near IL6	[26]	CT or CC—response	*p* = 0.010
UC	rs4696480	4	TLR2	[26]	TT—nonresponse	*p* = 0.040
UC	rs2569190	5	CD14	[26]	AG or AA—nonresponse	*p* = 0.040
UC	rs4251961	2	IL1RN	[26]	CT or CC—nonresponse	*p* = 0.040
CD	rs2274910	1	ITLN1	[30]	CC—response	*p* = 9.60 × 10^−3^
CD	rs13361189	5	IRGM	[30]	CC—response	*p* = 5.00 × 10^−3^
RA	rs10919563	1	PTPRC	[32]	A—nonresponse	*p* = 0.030
RA	rs12081765	1	intergenic	[33]	A—nonresponse	*p* = 7.39 × 10^−4^
RA	rs1532269	5	PDZD2	[33]	G—nonresponse	*p* = 7.37 × 10^−4^
RA	rs17301249	6	EYA4	[33]	C—response	*p* = 5.67 × 10^−5^
RA	rs7305646	12	intergenic	[33]	T—response	*p* = 1.47 × 10^−4^
RA	rs4694890	4	TEC	[33]	C—response	*p* = 6.47 × 10^−3^
RA	rs1350948	11	intergenic	[33]	A—nonresponse	*p* = 8.64 × 10^−3^
RA	rs7962316	12	LINC01619	[33]	G—nonresponse	*p* = 2.05 × 10^−2^
RA	rs4411591	18	LINC01387, LOC100130480	[6]	C—response	*p* = 5.14 × 10^−5^
RA	rs7767069	6	LOC102723883	[6]	A—nonresponse	*p* = 8.34 × 10^−5^
RA	rs4651370	1	near PLA2G4A	[6]	A—response	*p* = 1.09 × 10^−4^
RA	rs1813443	11	CNTN5	[6]	C—nonresponse	*p* = 1.37 × 10^−4^
RA	rs1447722	3	intergenic	[6]	C—response	*p* = 1.62 × 10^−4^
RA	rs1568885	7	intergenic	[6]	A—response	*p* = 1.69 × 10^−4^
RA	rs12142623	1	near PLA2G4A	[6]	A—response	*p* = 2.04 × 10^−4^
RA	rs2378945	14	NUBPL	[6]	A—nonresponse	*p* = 6.88 × 10^−4^
RA	rs6427528	1	CD84	[5]	GA or AA—response	*p* = 8.00 × 10^−8^
CD	rs4645983	1	CASP9	[10,34]	C—nonresponse	*p* = 0.040
CD	rs763110	1	FASLG	[34]	C—response	*p* = 0.002
CD	rs396991	1	FCGR3A	[35]	GG—response	*p* < 0.001
UC	rs1004819	1	IL23R	[36]	AA—response	OR = 1.27
UC	rs2201841	1	IL23R	[36]	GG—response	OR = 1.21
UC	rs10889677	1	IL23R	[36]	AA—response	OR = 1.26
UC	rs11209032	1	IL23R	[36]	AA—response	OR = 1.15
UC	rs1495965	1	IL23R	[36]	CC—response	OR = 1.29
UC	rs7517847	1	IL23R	[36]	GG—nonresponse	OR = 0.76
UC	rs10489629	1	IL23R	[36]	CC—nonresponse	OR = 0.81
UC	rs11465804	1	IL23R	[36]	GG—nonresponse	OR = 0.77
UC	rs1343151	1	IL23R	[36]	AA—nonresponse	OR = 0.85
CD	rs1061624	1	TNFRSF1B	[27]	A—T haplotype—nonresponse	*p* = 0.010
CD	rs3397	1	TNFRSF1B	[27]	A—T haplotype—nonresponse	*p* = 0.010
CD	rs10210302	2	ATG16L1	[10]	T—response	*p* = 8.11 × 10^−4^
CD	rs1143634	2	IL1B	[37]	C—nonresponse	*p* = 0.027
CD	rs909253	6	LTA	[38]	Homozygotes for the LTA NcoI-TNFc-aa13L-aa26 haplotype 1-1-1-1—nonresponse	*p* = 0.007

^1^ PA = psoriatic arthritis, RA = rheumatoid arthritis, IBD = inflammatory bowel disease, CD = Crohn’s disease, MS = multiple sclerosis, AS = ankylosing spondylitis, SpA = spondyloarthritis.

**Table 3 genes-14-00452-t003:** SNPs associated with either biological or clinical response to ADA treatment in CD patients.

Gene	SNP ID	Chr	*p* Value	Nonresponse Association Timeframe (Week)	Our Analyzed Nonresponse Genotype	Reported Nonresponse Genotype
IL17A	rs2275913	6	9.73 × 10^−3^	20	AA or AG	GG in RA, AA or AG in IBD
MIF	rs755622	22	4.00 × 10^−3^	4	CC or CG	G in AS
ARFGAP2	rs3740691	11	1.24 × 10^−5^	4	GG	A in AS
TLR2	rs4696480	4	1.48 × 10^−3^	20	AA	TT in UC
ICOSLG (B7RP1)	rs762421	21	2.11 × 10^−3^	4	AA or AG	AA in PED-IBD
CD14	rs2569190	5	8.00 × 10^−3^ 2.60 × 10^−2^	12 30	AA or AG AG or GG	AA or AG in UC
TNFa	rs361525	6	2.79 × 10^−3^	4	AA or AG	AG in IBD
CDKAL1	rs6908425	6	1.73 × 10^−2^	30	CT or TT	CT or TT in PED-IBD
TNFAIP3	rs6927172	6	3.87 × 10^−2^	4	CC	CG or GG in IBD
TNFRSF1A	rs767455	12	3.94 × 10^−2^	30	CC	CC in CD and RA
IL1RN	rs4251961	2	3.93 × 10^−2^	30	CC	CC or CT in UC
TNFa	rs1800629	6	6.15 × 10^−3^	4	GG	AA in CD, AS, SpA
TNFRSF1A	rs1800693	12	2.90 × 10^−2^	4	CC	C in MS
TNFRSF1B	rs1061622	1	3.25 × 10^−2^	12	GG	GG in AS
PTPRC	rs10919563	1	3.27 × 10^−2^	12	AA or AG	AA in RA
TLR2	rs11938228	4	9.82 × 10^−3^	20	CC	AA or AC in IBD
IRGM	rs13361189	5	4.45 × 10^−2^	12	CC or CT	TT in CD
TLR4	rs1554973	9	1.36 × 10^−2^	30	CC or CT	CC or CT in IBD
IL10	rs1800896	1	1.22 × 10^−2^	4	CC or CT	T in AS
ITLN1	rs2274910	1	4.79 × 10^−2^	4	CT or TT	TT in CD
IFNG	rs2430561	12	3.73 × 10^−2^	4	AA	A in RA TT in IBD
TLR9	rs352139	3	1.98 × 10^−2^	20	AA	AA in IBD
TRAF1	rs3761847	9	2.68 × 10^−2^	12	AA	GG in RA
PDE3A	rs3794271	12	4.40 × 10^−2^	4	TT	G in PA
TLR2	rs3804099	4	3.89 × 10^−2^	20	CC	TT in IBD
NLRP3	rs4612666	1	3.00 × 10^−2^	20	CC	T in RA
near IL1B	rs4848306	2	2.23 × 10^−2^	4	GG	AA in IBD
MAP3K14	rs7222094	17	3.59 × 10^−2^	4	CC	CT in IBD

## Data Availability

The data presented in this study are available on request from the corresponding author. The data are not publicly available due to patient privacy reasons.

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
