# Peer review of "MIF Variant rs755622 Is Associated with Severe Crohn’s Disease and Better Response to Anti-TNF Adalimumab Therapy"

_genes, 2023, doi:10.3390/genes14020452_

Round 1
Reviewer 1 Report (Previous Reviewer 3)
To improve the manuscript quality the author suggested responding to the following comments.
What is the reason to choose adalimumab (ADA) particularly?
ADA root of treatment?
Reason for choosing 2001 and 2015 publication?
reason for choosing 4, 12, 20 and 30 weeks for clinical and biological response
Total point of IBDQ?
What is the meaning of an IBDQ value higher than 170 points?
Table 1. Summary of pharmacogenetic anti-TNF signals - Disease location total numbers are not matching or need clarification.
Author Response
The reviewer wrote: To improve the manuscript quality the author suggested responding to the following comments.
What is the reason to choose adalimumab (ADA) particularly?
Reply: The first class of biological therapy indicated for inflammatory bowel disease was anti-TNF therapy. Two anti-TNF biological drugs are indicated for use in inflammatory bowel disease: infliximab (IFX) and adalimumab (ADA). In contrast to IFX, a chimeric monoclonal antibody, ADA is fully humanized and is appropriate for long-term anti-TNF therapy in inflammatory bowel disease. To this day, ADA remains the most common choice for biological therapy in inflammatory bowel disease. In addition, ADA is also indicated for other immune-mediated diseases such as psoriasis and rheumatoid arthritis. As such, there is a great deal of published information regarding ADA non-response due to its widespread use.
The reviewer wrote: ADA root of treatment?
The reply: ADA treatment is based on decreasing circulating TNF, both soluble and transmembrane TNF. TNF aka tumor necrosis factor alpha is an inflammatory cytokine with aberrant expression in many immune-mediated diseases. Targeting TNF allows decreases unnecessary inflammation and allows the inflamed tissue to heal, decreasing symptom severity or even inducing disease remission. Healing is driven mainly by a decrease of pro-inflammatory cell populations (CD4+ T cells and macrophages) and blocking intestinal epithelial cell death induced by TNFR1.
We have added following text to lines 36-40 to summarize anti-TNF drug action:
Targeting tumor necrosis factor α in IBD allows intestinal healing by blocking TNFR1-dependant intestinal epithelial cell death [2] and inducing cell death in macro-phages by binding to transmembrane TNF or by depriving TNFR2-dependent CD4+ T cell survival via NF-κB activation [3].
The reviewer wrote: Reason for choosing 2001 and 2015 publication?
Reply: In 2016, the first adalimumab biosimilar has been approved by the FDA in the US. Although these biosimilar drugs are demonstrated to function identically to their originator, we nevertheless wished to limit our data to the adalimumab originator drug only. Hence our decision to only choose publication to up but not including 2016.
The reviewer wrote: reason for choosing 4, 12, 20 and 30 weeks for clinical and biological response
Reply: The study design where clinical data was obtained had patient visits at 4, 12, 20 and 30 weeks when they could fill out the questionnaire and have blood testing done. Multiple timepoints were selected to also study time-dependence of ADA non-response.
The reviewer wrote: Total point of IBDQ?
Reply: IBDQ scores range from 32 and 224.
IBDQ has an advantage over other disease severity scoring algorithms (such as CDAI) by also estimating more of the patient’s subjective experience.
The reviewer wrote: What is the meaning of an IBDQ value higher than 170 points?
Reply: An IBDQ score of 170 points or higher represents moderately high patient quality of life, which has been defined in clinical praxis as a meaningful therapeutic goal (Hlavaty T. et al., Evaluation of short-term responsiveness and cutoff values of inflammatory bowel disease questionnaire in Crohn's disease. Inflamm Bowel Dis 2006, 12, 199-204.).
The reviewer wrote: Table 1. Summary of pharmacogenetic anti-TNF signals - Disease location total numbers are not matching or need clarification.
Reply: Please note that L4 location according to Montreal classification is not a location per se, but a location modifier used to describe presence of symptoms in the upper gastrointestinal tract in addition to the primary disease location (L1, L2 or L3). For example, a patient may be described as L1+L4, L2+L4 or L3+L4.
Reviewer 2 Report (New Reviewer)
IT is a useful study for the treatment of the patiens with CD and no response in anti-TNF therapy.
Author Response
Thank you kindly for the comment.
Reviewer 3 Report (New Reviewer)
Minor marks:|
1. Background, I would suggest providing more information about the influence of anti-TNF drug and their effect on the signaling pathways clinically crucial for Crohn’s disease.
2. What were the exclusion criteria for this research? Additionally, authors should provide a table with the clinical parameters, such as CRP, etc., for included patients.
3. Did the authors consider creating a regression model including the occurrence of the selected gene variant together with clinical parameters for further estimation of clinical respondence or non-respondence?
Author Response
The reviewer wrote: Minor marks:|
1. Background, I would suggest providing more information about the influence of anti-TNF drug and their effect on the signaling pathways clinically crucial for Crohn’s disease.
Reply: We have added following text to lines 36-40 to summarize anti-TNF drug action:
Targeting tumor necrosis factor α in IBD allows intestinal healing by blocking TNFR1-dependant intestinal epithelial cell death [2] and inducing cell death in macro-phages by binding to transmembrane TNF or by depriving TNFR2-dependent CD4+ T cell survival via NF-κB activation [3].
The reviewer wrote: 2. What were the exclusion criteria for this research? Additionally, authors should provide a table with the clinical parameters, such as CRP, etc., for included patients.
Reply: Exclusion criteria were as described previously in reference 10, Koder et al. Pharmacogenomics 2015. The following text has been added to the manuscript to list the previous exclusion criteria:
Exclusion criteria were: other complications of CD (e.g., stenosis, abscesses, previous total colectomy), a history of allergy to murine proteins, a serious infection in the previous 3 months, positive test on tuberculosis or active tuberculosis, malignancy, pregnancy and lactation)
Table 1 has also been appropriately extended with available clinical data.
The reviewer wrote: 3. Did the authors consider creating a regression model including the occurrence of the selected gene variant together with clinical parameters for further estimation of clinical respondence or non-respondence?
Reply: Regression models were considered. However, testing for assumptions required for the regression analysis showed a VIF greater than 50 and a conditional index greater than 30, suggesting strong multicollinearity. As such, regression analysis was not performed as it is likely to present inaccurate results.
This manuscript is a resubmission of an earlier submission. The following is a list of the peer review reports and author responses from that submission.
Round 1
Reviewer 1 Report
The reading of the manuscript is not easy, the presentation of the data and how they were collected is not clear.
The interesting field of pharmacogenetics requires rigorous protocols in order to guarantee that the reader is not misled in the interpretation of the data, but the scientific evidence and conclusions, in this case, are confounding.
In line 34, the abbreviation NSAIDs is not shown, and also in lines 39 for IBD and 40 for AS, in line 58 for UC.
In line 50-52 the aim of the study is not clearly described, please review the sentence.
The patients enrolled are not well presented and also the clinical parameters taking into account for the analysis.
For the genetic analysis, 41 SNPs are selected, but the methodology performed to genotyping all the CD patients is not clear.
I think that using the available functional and biological databases was not useful instead of a better characterization of the variants described in association with the therapeutic response.
It is not clear how the 22 studies taken in consideration can be useful considering the cohort of patients under analysis.
The table 1 reports the data in an unclear way, some lines are not complete and the genotype relating to the response to the therapy is not understandable.
The caption describing the table it is not clear, and following is reported “2SNP already analyzed in our cohort” that are reported only in 4 variants not discussed in the manuscript.
About the pharmacogenetics analysis, no data supporting the consistent association in the cohort of patients and the treatment are present. The variants taken into consideration are not well described (HGSV nomenclature, frequency in population).
Moreover, I don't understand why the genotype that might predispose to a different response to treatment should vary over time, and is reported in figure 1 only for SNP rs755622 in MIF gene.
Table 2 introducing SNPs associated with either biological or clinical response to ADA treatment in CD patients is not clear.
Also gene ontology analysis, the discussion and the conclusion reported are not consistent or clearly presented.
Author Response
The reviewer wrote:
The reading of the manuscript is not easy, the presentation of the data and how they were collected is not clear.
The interesting field of pharmacogenetics requires rigorous protocols in order to guarantee that the reader is not misled in the interpretation of the data, but the scientific evidence and conclusions, in this case, are confounding.
In line 34, the abbreviation NSAIDs is not shown, and also in lines 39 for IBD and 40 for AS, in line 58 for UC.
Our reply:
The abbreviations have been corrected accordingly.
The reviewer wrote:
In line 50-52 the aim of the study is not clearly described, please review the sentence.
Our reply:
Line 50-52 has been ameliorated for clarity of the study’s aim.
The reviewer wrote:
The patients enrolled are not well presented and also the clinical parameters taking into account for the analysis.
Our reply:
We have added an additional table (Table 1) displaying clinical data for our cohort.
The reviewer wrote:
For the genetic analysis, 41 SNPs are selected, but the methodology performed to genotyping all the CD patients is not clear.
Our reply:
Text explaining the methodology and with an appropriate citation has been added to the manuscript in lines 82-83.
The reviewer wrote:
I think that using the available functional and biological databases was not useful instead of a better characterization of the variants described in association with the therapeutic response.
It is not clear how the 22 studies taken in consideration can be useful considering the cohort of patients under analysis.
Our reply:
In this study, we gathered 22 studies with anti-TNF response markers reported for different immune-mediated diseases treated with anti-TNF agents. To this date, anti-TNF response markers have not yet been translated into clinical praxis from a lack of robust replication between cohorts. Thus, our study also aimed to replicate previously published anti-TNF response markers in different immune-mediated diseased treated with ADA in our cohort of CD patients.
Unfortunately, functional analysis of variants associated with anti-TNF response is not within the scope of our study beyond gene ontology analysis of genes containing anti-TNF response variants.
The reviewer wrote:
The table 1 reports the data in an unclear way, some lines are not complete and the genotype relating to the response to the therapy is not understandable.
Our reply:
Data missing in row 4 has been added. The font has been decreased slightly and column width has been optimized for better clarity.
The reviewer wrote:
The caption describing the table it is not clear, and following is reported “2SNP already analyzed in our cohort” that are reported only in 4 variants not discussed in the manuscript.
Our reply:
The line does not contribute any meaningful information and has been removed.
The reviewer wrote:
About the pharmacogenetics analysis, no data supporting the consistent association in the cohort of patients and the treatment are present. The variants taken into consideration are not well described (HGSV nomenclature, frequency in population).
Our reply:
Due to the constraints of the table in the manuscript, a supplementary table (Table S2) is provided with HGSV nomenclature and population frequency in the dbSNP European cohort, which is the closest approximation to the investigated Slovenian cohort.
The reviewer wrote:
Moreover, I don't understand why the genotype that might predispose to a different response to treatment should vary over time, and is reported in figure 1 only for SNP rs755622 in MIF gene.
Our reply:
Clinical and biological response may vary over time due to mainly due to primary and secondary non-response. Primary non-response is believed to be a lack of response during therapy induction, possibly due to a different molecular pathology of the patients, such as primarily interleukin-driven inflammation instead of TNF-driven inflammation. After therapy induction, patients may also lose response over time due to various factors, such as the formation of antibodies against the anti-TNF drug or the pathological processes compensating for the lack of TNF by activating other molecular pathways of inflammation.
However, in the case of rs755622, response does not vary over time in relation to genotype. Instead, the pre-treatment IBDQ scores only suggest that the GG genotype is associated with more severe disease, but later timeframes indicate that the GG genotype is also associated with very favorable anti-TNF response and IBDQ improvement.
Additional figures have been added for SNP rs3740691 and rs2275913. Both figures demonstrate a stable response over 30 weeks.
The reviewer wrote:
Table 2 introducing SNPs associated with either biological or clinical response to ADA treatment in CD patients is not clear.
Our reply:
The table’s header has been changed for clarity. Column width has been optimized.
The reviewer wrote:
Also gene ontology analysis, the discussion and the conclusion reported are not consistent or clearly presented.
Our reply:
Additional text addressing the gene ontology results in the context of the study has been added to the Discussion (lines 258-272).
Reviewer 2 Report
Comments and suggestions for authors:
The authors aimed to find the newly markers associated with response to adalimumab in Crohn’s disease patients by using genetic analysis.
First, based upon SNP selection from literature research, they showed that three novel pharmacologic association including rs755622 encoding MIF, rs3740691 encoding ARFGAP2 and rs2275913 encoding IL17A. Finally, they also demonstrated that the variant rs2275913 encoding IL17A is the strongest and most consistent association with response to adalimumab. The author's findings are very interesting; however, I have several concerns.
Major concerns:
1. As they mentioned in Discussion, why is the variant with IL-17A associated with strongly response to adalimumab in CD patients ? It is better to describe the role of IL-17 for inflammatory bowel disease including CD. For example, it has been reported that IL-17 regulates intestinal epithelial permeability (Lee et al. Immunity 2015). As related to above, the pathological mechanism of MIF is needed to describe in Discussion.
2. At the last paragraph in Discussion, they need to add the sentences about underlying mechanism between NF-kB signaling and bacterium response pathways in CD patients.
Minor concerns:
1. In Discussion, line 211: They described “IL-17A, which is produced mainly by Th1 and Th1/Th17 cells, ..” ; however, Th1 cells are not main source of IL-17A.
2. English should be revised through all sentences by native speaker’s check.
Author Response
REVIEWER 2
The reviewer wrote:
The authors aimed to find the newly markers associated with response to adalimumab in Crohn’s disease patients by using genetic analysis.
First, based upon SNP selection from literature research, they showed that three novel pharmacologic association including rs755622 encoding MIF, rs3740691 encoding ARFGAP2 and rs2275913 encoding IL17A. Finally, they also demonstrated that the variant rs2275913 encoding IL17A is the strongest and most consistent association with response to adalimumab. The author's findings are very interesting; however, I have several concerns.
Major concerns:
1. As they mentioned in Discussion, why is the variant with IL-17A associated with strongly response to adalimumab in CD patients ? It is better to describe the role of IL-17 for inflammatory bowel disease including CD. For example, it has been reported that IL-17 regulates intestinal epithelial permeability (Lee et al. Immunity 2015). As related to above, the pathological mechanism of MIF is needed to describe in Discussion.
Our reply:
We thank the reviewer for this reference. Text briefly explaining IL-17 and MIF in pathogenesis and anti-TNF response in CD has been added to lines 227-233.
The reviewer wrote:
2. At the last paragraph in Discussion, they need to add the sentences about underlying mechanism between NF-kB signaling and bacterium response pathways in CD patients.
Our reply:
Additional text addressing the gene ontology results in the context of the study has been added to the Discussion (lines 267-281).
In addition, the newly provided Figure 5 should illustrate key pathways.
The reviewer wrote:
Minor concerns:
1. In Discussion, line 211: They described “IL-17A, which is produced mainly by Th1 and Th1/Th17 cells, ..” ; however, Th1 cells are not main source of IL-17A.
Our reply:
This oversight was corrected. The sentence now correctly states the following:
“IL-17A, which is produced mainly by Th17 cells,…”
The reviewer wrote:
2. English should be revised through all sentences by native speaker’s check.
Our reply:
The text has been reviewed by a native speaker.

Reviewer 3 Report
Author explained clearly, so no comments.
Author Response
Thank you.